# Left-Behind Experiences and Cyberbullying Behavior in Chinese College Students: The Mediation of Sense of Security and the Moderation of Gender

**DOI:** 10.3390/bs13121001

**Published:** 2023-12-07

**Authors:** Haiying Wang, Shuang Wu, Weichen Wang, Yuming Xiao

**Affiliations:** 1School of Psychology, Northeast Normal University, Changchun 130024, China; wangwc036@nenu.edu.cn (W.W.); xiaoym329@nenu.edu.cn (Y.X.); 2Guidance and Service Center for Students, Northeastern University, Shenyang 110819, China; wushuang@mail.neu.edu.cn

**Keywords:** cyberbullying behavior, left-behind experiences, sense of security, gender, moderated mediating effect

## Abstract

Left-behind children seem to be more sensitive in interpersonal communication, find it more difficult to establish a stable, safe relationship with surrounding people, and have fewer positive coping styles when encountering problems, thus the aim of the present study was to explore the association between left-behind experiences and cyberbullying behavior among Chinese college students through the mediation of sense of security and the moderation of gender. A questionnaire survey comprised 553 college students with left-behind experiences and 526 college students without such experiences. The results showed that, firstly, cyberbullying behavior was significantly higher in college students with left-behind experiences than those without such experiences; secondly, left-behind experiences and cyberbullying behavior in college students was partially mediated by a sense of security; and finally, that gender moderated the mediation of the sense of security between left-behind experiences and cyberbullying behavior. This study suggests the family environment is important for individual growth and illustrates how the influence of childhood left-behind experience persists in individuals.

## 1. Introduction

### 1.1. College Students with Left-Behind Experiences

Labor migration has been a global trend that changes the family structure and stability [1]. Since the 1980s, the floating population in China has expanded massively alongside accelerated economic growth and urbanization. To reduce living costs and avoid high education burdens, migrant workers may choose not to relocate their children with them [2]. This phenomenon leads to the formation of a special group, the left-behind children (LBC). Currently, there are many college students who have childhood left-behind experiences (LBEs). College students with LBEs are defined as those who are enrolled in school and where one or both parents work away from home for over six months before the students turn 16 [3]. According to the survey of Liu and Wang [4], the percentage of college students with LBEs in China is 46.92%. Chinese college students with LBEs also exhibit lower self-esteem, self-efficacy, and mental health issues [5,6,7,8]. Therefore, an investigation into the impact of LBEs on adulthood is necessary. Findings would provide insight into the long-term effects of LBEs on an individual’s psychological development and assist in developing guiding strategies to help Chinese college students with LBEs.

### 1.2. Left-Behind Experiences and Cyberbullying Behavior

With the widespread use of the internet and multiple communication technologies, the traditional learning and communication environment has gradually shifted to online networks, for which college students experience new feelings, however this can have negative effects. Cyberbullying (CB), which is an intentional, repetitive, and aggressive behavior committed by individuals or groups against other individuals using information and communication technologies, has emerged and continues to increase as a new form of bullying [9,10,11]. Perpetrators of CB are prone to depression and anxiety [12], as well as dropping out of school [13]. The victims are also vulnerable to psychological problems such as low self-esteem, suicidal ideation, and in some cases physical symptoms such as headache and insomnia [14,15]. Thus, scrutinizing the internal mechanism of CB is of great importance for prevention and intervention in addition to promoting the physical and mental health of college students.

According to the General Aggression Model (GAM), the influences of CB include situational and individual factors. The former encompasses the school climate and parental involvement, and the latter includes personality, motivation, values, and other psychological characteristics [16]. Adolescents with a high level of bullying exhibit weaker emotional ties to their parents and lack effective parental supervision and support [17,18,19]. Similarly, parental involvement and sound family management (e.g., parental regulation) are negatively associated with CB [20,21]. In addition, parental care correlates negatively with aggression [22] through positive emotional support from the family as a protective factor against individual involvement in CB [23,24]. Previous results have shown that college students with LBEs are more aggressive [25,26] and have a higher incidence of problematic behavior compared to college students without LBEs [27]. According to attachment theory, children who were separated from their parents during childhood are unable to establish safe, emotional connections with their parents, displaying an unsafe attachment relationship; as a result, they tend to be more sensitive to interpersonal communications and experience more negative emotions. For these children, it is difficult to establish a stable, safe relationship with people around them, and they possess fewer positive coping styles when encountering problems [28]. They are more likely to externalize their emotions and show some aspects of behavioral problems [29]. Luo et al. [30] concluded that individuals with LBEs are estranged from their parents, lack parental role models, and have lower family intimacy, and most have experienced negative parenting styles, such as punishment and interference. These factors all increase the risk of CB among college students with LBEs. Therefore, exploring the relationship between LEBs and CB is necessary.

### 1.3. The Mediation of Sense of Security

Sense of security refers to an individual’s premonition of possible physical or psychological harm, in addition to the individual’s sense of strength/powerlessness to deal with it, which is primarily a sense of certainty and control [31]. The formation and development of a sense of security relates significantly to the family environment [32,33].

Due to the separation from their parents in childhood, college students with LBEs suffer from a lack of parental care and companionship, they communicate less with their parents, and have difficulty in forming secure relationships, which often results in security deficits. Following this, the individual’s insecurity may aggravates over time and this affects their sense of security level in adulthood [34]. College students with LBEs have a lower sense of security than those without an LBE [26]. According to the hierarchical theory of needs, when an individual’s sense of security is not satisfied, they perceive everything as dangerous, distrust people and the things around them, and may externalize their unsafe psychology into problematic behaviors [35]. An [36] showed that the sense of security of college students with LBEs is significantly and negatively correlated to network problematic behaviors. When comparing the family and school environments, effective supervision and restraint are notably absent in the network setting. With a lower sense of security following separation from their parents, college students with LBEs may be more willing to seek psychological release via the internet which exposes personality-related tendencies that achieve psychological satisfaction through CB.

### 1.4. The Moderation of Gender

Some scholars have studied gender differences in CB, but there are different views. One perspective is that there is no gender difference in college students’ CB, for example, Slonje and Smith [37] found almost no gender difference in college students’ CB. Jin et al. [38] revealed no significant gender difference in the reactive and instrumental attacks of CB. However, there is another view that states there are substantial gender differences in college students’ CB. Males are more aggressive than females, making them more likely to attack others and become cyberbullies [39,40,41]. However, Gorzig and Olafsson et al. [42] found females to be more active in CB. The gender difference in CB is currently a controversial issue and further exploration into the role of some variables in gender differences in CB is necessary. According to differences in social and cultural expectations, males are generally strong and independent, whereas females are gentle and tactful. Females are more inclined to communicate with their parents when encountering new experiences, and they have a greater need of their parents’ companionship and warmth during childhood. In one study, the attachment anxiety and attachment avoidance of left-behind female college students were significantly higher than those of left-behind male college students [43]. Females growing up in a left-behind circumstance cannot communicate or resolve new problems with their parents when they encounter them due to the lack of one, or both, parents’ companionship from a young age, which made it difficult for them to form a safe parent-child attachment relationship. Whether there will eventually be a phenomenon of lower sense of security than boys needs to be further explored. Gender moderates the relationship between psychological abuse in childhood and CB in adolescents [44]. In addition, gender moderates the relationship between family environment complexity and a child’s interpersonal security, and moreover, high family environment complexity has a greater negative impact on a girl’s interpersonal security [45].

### 1.5. The Current Study

In order to investigate the long-term adverse effects of LBEs and how it affects college students’ CB, this study adopted a cross-sectional approach based on the General Aggression Model and attachment theory with the aim of exploring the relationship between LBEs and CB, as well as the mediating effect of sense of security and the moderating effect of gender. This study hypothesizes that (Figure 1):

 **Hypothesis 1.** *College students’ LBEs positively associated with CB*.

 **Hypothesis 2.** *A sense of security mediates the relationship between college students’ LBEs and CB*.

 **Hypothesis 3.** *Gender moderates the relationships between college students’ LBEs and their sense of security and, furthermore, between LBEs and CB*.

## 2. Materials and Methods

### 2.1. Participants

In this study, 1209 students were recruited through anonymous online questionnaires, predominantly from universities in Changchun, Harbin, Shenyang, Qingdao, Hangzhou, and other cities in China. The subjects’ regular answer data were excluded, and 1079 valid data points were obtained. The effective recovery rate of the questionnaire was 89.25% and the average age of the subjects was 20.17 ± 1.86 years. The subjects included 553 (51.25%) with LBEs, and 526 (48.75%) without LBEs; 455 were males (42.17%), and 624 were females (57.83%). The participants all had normal visual acuity and no mental illness. The study was approved by the Academic Ethics Committee of the College of Psychology of Northeast Normal University.

### 2.2. Measures

#### 2.2.1. Cyberbullying Scale

The Cyberbullying Scale compiled by Erdur and Kavsut [46] and revised by Zhou et al. [47] has favorable reliability and validity and has been widely used in studies of CB amongst college students. The scale includes a cyberbullying subscale and a victim-bullying subscale, with 18 items for each. This study used the cyberbullying subscale to measure the extent of bullying others in the online environment (e.g., ‘I have insulted someone in a chatroom’). Participants were asked to report their experiences with 4-point Likert-type, providing a score for having experienced cyberbullying in the past year (where 1 = ‘it has never happened to me’, 2 = ‘it happened once or twice’, 3 = ‘it happened three to five times’, and 4 = ‘it happened more than five times’). The higher the score, the higher the levels of individuals bullying others online. In this study, the subscale had favorable validity (KMO = 0.94, *p* < 0.001) and reliability, with a Cronbach’s alpha of 0.92.

#### 2.2.2. Left-Behind Experiences

LBEs was evaluated using a questionnaire designed by the authors which included gender, student cadre, and LBE. According to the definition of LBEs supplied by Zuo et al. [3], the question of LBEs was “Between your birth and your turning 16 years of age, did one or both of your parents go away to work for more than six months ? (0 = no, 1 = yes)”.

#### 2.2.3. Sense of Security Questionnaire

This study used the Sense of Security Questionnaire based on Chinese groups developed by Cong and An [48]. According to the concept of sense of security and its theoretical framework, Cong and An [48] selected the Maslow Safety-Unsafe Scale to test criterion validity and the results showed a significant correlation between the two (*r* = −0.68, *p* < 0.01). Due to its favorable reliability and validity, the questionnaire has been widely used in recent studies of Chinese college students. The scale consists of 16 items, including 2 dimensions, the first of which is interpersonal security (8 items, e.g., ‘People say I am a shy person’) and the second is personal sense of certainty and control (8 items, e.g., ‘I feel incapable of dealing with sudden danger’). Interpersonal security reflects mostly on the individual’s safety experience during interpersonal interactions, whereas personal sense of certainty and control reflects the individual’s prediction of life and their sense of certainty and control. Using a 5-point Likert scale, ranging from very consistent to very inconsistent, the responses were recorded as 1, 2, 3, 4, and 5; the higher the total score, the greater the individual’s sense of security. In this study, the questionnaire had favorable validity (KMO = 0.93, *p* < 0.001) and reliability with a Cronbach’s alpha of 0.90.

### 2.3. Procedure and Data Analyses

Undergraduates from multiple universities agreed to participate in the online survey. This survey could only be submitted if all of the were completed. The questionnaires were anonymous, and there were no correct or incorrect answers. Participants were asked to answer according to their actual situation and informed of their right to withdraw at any time. SPSS21.0 and Hayes’ [49] SPSS macro PROCESS were used to collate and analyze the data. First, the incidence of CB among college students with or without LBEs was calculated based on whether one or more CB events had occurred. Second, variance analysis was employed to test the differences in CB scores (exposure or involvement in CB events) according to gender, grade, and whether they had been student cadres. Third, an independently sampled *t*-test was used to examine the differences in LBEs on CB and sense of security among the participants. Finally, dummy variables were used to deal with whether there was a LBE. Model 8 of PROCESS was used to test the mediating effect of sense of security and the moderating effect of gender on the first half of the mediating effect and the direct effect.

## 3. Results

### 3.1. Common Method Variance Control and Test

As this study collected data through a self-reported method, a common method variance (CMV) issue may exist. According to the suggestion by Zhou & Long [50], in the data collection stage, the participants were told that the results would be kept anonymous so as to reduce this possible deviation. After the data collection was complete, Harman’s one-factor test was harnessed to detect CMV [51]. There were 5 factors with eigenvalues greater than 1, and the first factor to explain the variance accounted for 28.29%, which was less than the critical value of 40%. Consequently, there was no significant CMV in this study.

### 3.2. Demographic Characteristics of the College Students in Relation to CB

The incidence of CB amongst college students with LBEs is 46.47%; amongst college students without LBEs, that figure is 34.22%. The outcome of the independent sample *t*-test showed that CB was significantly different in the demographic variables of gender (*t* = −4.63, *p* < 0.001) and student cadre (*t* = 3.89, *p* < 0.001). The scores of males (*M* = 20.28, *SD* = 4.70) were significantly higher than those of females (*M* = 19.04, *SD* = 3.75). The CB scores of student cadres (*M* = 20.10, *SD* = 5.34) were significantly higher than those of non-student cadres (*M* = 19.08, *SD* = 2.76).

### 3.3. Analysis of the Differences in CB and Sense of Security among College Students with LBEs

An independent sample *t*-test was conducted on the CB of college students with or without LBEs. The total average CB score of college students with LBEs was 20.07 and the standard deviation was 5.04. The total average CB score of college students without LBEs was 19.03 and the standard deviation was 3.04. The CB score of college students with LBEs was significantly higher than those without LBEs (*t* = −4.10, *p* < 0.001).

Independent samples *t*-tests were conducted on the sense of security of college students with or without LBEs. The total average score of college students with LBEs was 41.46 and the standard deviation was 11.68. The average score of college students without LBEs was 46.39 and the standard deviation was 11.07. The score of college students with LBEs was significantly lower than that of college students without LBEs (*t* = 7.11, *p* < 0.001).

### 3.4. Descriptive Statistics and Correlation Analysis of Variables

Table 1 shows the average, standard deviation, and correlation matrix of each research variable. The results indicate that LBEs are significantly correlated to CB, and that sense of security is negatively correlated to CB. In addition, to avoid the influence of student cadres, it was treated as a control variable in the subsequent regression analysis.

### 3.5. Moderated Mediation Analysis

This study used the PROCESS in Model 8 developed by Hayes to test the mediating effect of sense of security and the moderating effect of gender. Under the condition of controlling student cadres, the LBE was taken as the independent variable, CB was taken as the dependent variable, sense of security was taken as the mediator, and gender was taken as the moderator. All variables were standardized and brought into the regression equation.

The results are shown in Table 2. The study found that LBEs were significantly associated with sense of security in Model 1 and that LBEs and sense of security were significantly associated with CB in Model 2. The bootstrap method of deviation correction was further used to test the mediating effect, as presented in Table 3. Therefore, sense of security is partially mediated between college students’ LBEs and CB.

The interaction between LBEs and gender were significantly associated with sense of security (*β* = 0.62, *t* = 5.18, *p* < 0.001) but did not associate significantly with CB (*β* = 0.13, *t* = 1.47, *p* > 0.05). The results indicated that gender moderated the relationship between LBEs and sense of security. The mediating effect of gender-specific college students’ sense of security between LBEs and CB, and its 95% bootstrap confidence interval, are depicted in Table 4.

To further investigate the moderating effect of gender on the relationship between LBEs and sense of security, a simple slope analysis was conducted (see Figure 2). For female college students, LBEs formed a significant negative association with their sense of security (*b_simple_* = −0.67, *t* = −8.99, *p* < 0.001), suggesting that for female college students, LBEs will significantly reduce their sense of security. The impact of LBEs on male college students’ sense of security declined but was not significant (*b_simple_* = −0.05, *t* = −0.52, *p* > 0.05).

## 4. Discussion

### 4.1. Demographic Characteristics of College Students’ CB

In the numerous studies relating to CB, gender issues have caused controversy. However, the findings of this study support the conclusion that CB is higher in males than females. On the one hand, this could be attributed to the fact that males gained earlier internet access than females and spend more time surfing the internet on a daily basis [52]. On the other hand, females are generally more cautious in their use of the web than males [53]. Therefore, the context and conditions for CB is more favorable for males than females.

This study found that student cadres have significantly higher CB scores than non-student cadres, which is similar to the findings of studies into aggressive tendencies among college students [54]. Their higher involvement in CB may be related to the higher workload, additional pressures, and strong competitiveness that comes with being a student cadre [55,56]. When confronted with questioning and misunderstandings from classmates, they are more inclined to express emotions such as sadness and anger in the online environment, thus causing harm to others [57]. In addition, it reminds us to be more attentive towards the emotional and psychological health of student cadres.

### 4.2. The Relationship between College Students’ LBEs and CB

This study explored the relationship between college students’ LBEs and CB. College students with LBEs have significantly higher CB scores than those without, which is consistent with a previous conclusion that college students with LBEs have more problematic behaviors and higher aggression [25,26,27]. This outcome supports Hypothesis 1 and points to an important link between college students’ LBEs and CB. The family function of LBC, whose parents left home to work, is relatively weak [58]; family cohesion and parental support are weak, and their children perceive less family warmth [22,59]. College students with LBEs grew up in an adverse environment containing a relative lack of affection, making them prone to external problems. Depressed emotions in real life are brought into the unfettered network environment, thereby facilitating CB. In addition, the positive parenting style of college students with LBEs is significantly lower than that of college students without LBEs [22,59]. Individuals with a negative parenting style are more likely to have psychological and behavioral problems and to harm themselves or others [4]. This conclusion also verified the impact of family environmental factors on individual behaviors in the GAM [13]. A low-income family structure, negative parenting styles, and fewer parent-child interactions may lead to CB [60]. Hence, college students with LBEs have higher CB scores than those without LBEs in environments with unstable family structures, low family intimacy, and less parent-child communication.

### 4.3. The Mediation of Sense of Security

Based on existing research, this study also found that the sense of security partially mediated the relationship between college students’ LBEs and CB. That is, college students’ LBEs had a direct effect on the occurrence of CB or had an indirect association with the occurrence of CB by reducing sense of security. Hypothesis 2 was verified. College students with LBEs had a lower sense of security than those without LBEs, which is consistent with the conclusion of Zhang and Xu [26]. Attachment theory points out that an individual’s sense of security is closely tied to their upbringing and their parent-child relationships [61]. The hierarchy of needs theory asserts that a stable and harmonious growth environment can meet an individual’s need for a sense of security [35]. Emotional ties between family members can increase family cohesion, enable individuals to feel supported by family members, and contribute to the formation of an individual’s sense of security [62]. College students with LBEs tend to feel less secure compared to those without LBEs. This may be due to the lack of parental care during their childhood due to prolonged separation from their parent(s) and low family intimacy which hinders their ability to establish stable and safe attachment relationships with their parents. Moreover, this attachment model, which is established with their parents in childhood, will affect how they interact with others in adulthood; they will encounter more difficulties and problems during interpersonal communications and feel a lower sense of security.

The finding that states the lower the college student’s sense of security, the more they engage in CB is not in accordance with Al Qudah et al. [63]. They concluded that one’s sense of security does not significantly predict CB. This may be because some participants selected in this study suffered from LBEs had a lower sense of security compared with the average level, whereas Al Qudah et al. [63] investigated all college students. In addition, different research tools may lead to different numbers of CB being reported by the participants, which makes the relationship between sense of security and CB inconsistent. However, this finding from the present study aligns with other research on sense of security and problematic behaviors [26]. A lower sense of security can cause pessimism, inferiority, sensitivity, hostility to others, and egocentrism, which affect problematic behaviors [64]. On the one hand, individuals with a low sense of security have negative attitudes of doubt or distrust toward the external environment and others, so they will find it difficult to communicate and express themselves effectively with others [31]. Thus, it causes them to feel a lack of control and diminishes their actual experiences of interpersonal relationships and their ability to predict life events [48]. To make up for this sense of control from the network environment, they generated cyberbullying. On the other hand, individuals with a low sense of security cannot effectively assess and predict external risks, tending to choose negative coping styles [65] and thus exhibit more CB. College students with LBEs usually experience low-income family functioning in childhood and they display poor communication among family members, loose emotional connections, and inadequate security attachment [60,66]. After long exposure to a negative family environment, they progressively internalize their sense of insecurity as a negative emotional state. They become sensitive, suspicious, distrustful of surrounding people, and prone to pursuing negative behaviors, which in turn leads to CB.

### 4.4. The Moderation of Gender

This study also found that gender moderated the relationship between LBEs and sense of security, which verified the latter part of Hypothesis 3. Specifically, for female college students there was a significant and negative correlation between LBEs and sense of security, but this relationship was not significant among male college students. This may be because the family environment—when separated from their parents—causes females to become more sensitive [45,67]. Female gender characteristics make them exhibit a higher demand for a stable living environment and parental support [68]. The lack of parental presence and care make females more susceptible to unsafe attachments, thereby reducing their sense of security [45,67]. According to the social and cultural model, social and cultural factors directly contribute to gender differences in personality traits. Males and females are expected to develop different personality traits to adapt to different social roles [69]. Males are perceived to be stronger and braver than females. With an incomplete family environment and the lack of effective parental supervision, the strong personality traits of males make them take on greater family responsibilities. Through difficult training, they become stronger and braver. Therefore, LBEs have no significant effect on the reduction of their sense of security. Unlike the left-behind males, left-behind females need more attention from their parents to improve their sense of security and increase the courage they need to face negative life events.

However, the results of this study imply that gender has no significant moderating effect on the direct relationship between LBEs and CB. The first half of Hypothesis 3 was not verified, suggesting that the relationship between LBEs and CB may have cross-gender stability. It may be due to the high sociocultural tolerance for male problematic behavior, however, the prevalence of male problems remains higher than that of female problems in college students with LBEs [27]. Female college students with LBEs have a higher sense of inferiority than male college students with LBEs [70]. Individuals with a high sense of inferiority tend to exhibit more aggression [26]. In the context of a left-behind upbringing, both males and females face their own difficulties, which can easily lead to problematic behaviors. Therefore, the direct relationship between LBE and CB may not be moderated by gender.

### 4.5. Limitations

Based on the GAM, this study explored the relationship between college students’ LBEs and CB and its internal mechanisms. The results showed that college students’ LBE was related to CB via sense of security and that this relationship was moderated by gender. This study made a novel contribution to the empirical literature on the long-term effects of childhood LBEs. Using data from Chinese college students, we found that experiences were negatively associated with mental health and behaviors in late adolescents. This not only underscores the importance of the family environment on individual development but also suggests that LBE in childhood is a risk factor for individuals. Therefore, we should pay attention to college students with LBEs and provide them with more support and the appropriate behavioral guidance.

Although this study makes certain contributions to research on college students with LBEs, several limitations should be mentioned. Firstly, this study did not further refine LBEs. Future research could refine and classify the demographic variables of LBEs and explore the influence of different types of LBEs on the psychology and behavior of college students. Secondly, the various types of bullying behavior have not been compared and examined, which may also be one of the main reasons affecting the results. Future studies should continue to explore the moderating effect of gender. Thirdly, due to spatial and temporal constraints, this study has some limitations regarding using a cross-sectional research design to explain the internal causality. In future, a longitudinal approach should be used to explore the relationships between variables, and further reveal the longitudinal mediating role of sense of security in the relationship between LBEs and CB.

## 5. Conclusions

This study found that 46.47% of college students had LBEs, indicating that having LBEs are a common phenomenon among Chinese college students. LBEs are an environmental factor that influence CB, and the relationship between the two is mediated by a sense of security, and in addition, gender mediated the relationship between the two. Furthermore, when compared to male college students, the mediating effect was significant only in female college students. In the future, researchers and teachers should pay more attention to college students’ sense of security, so as to reduce the incidence of CB among them and alleviate the adverse effects of LBEs on college students.

## Figures and Tables

**Figure 1 behavsci-13-01001-f001:**
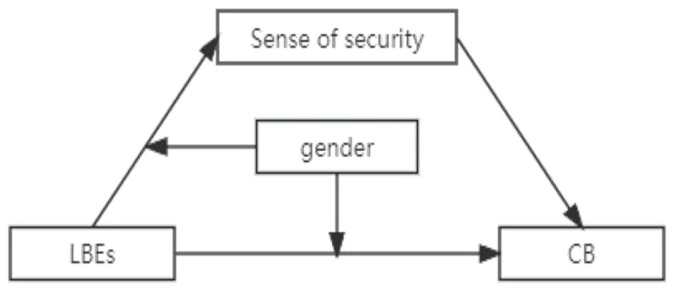
The moderated mediation effect among LBEs, CB, sense of security, and gender.

**Figure 2 behavsci-13-01001-f002:**
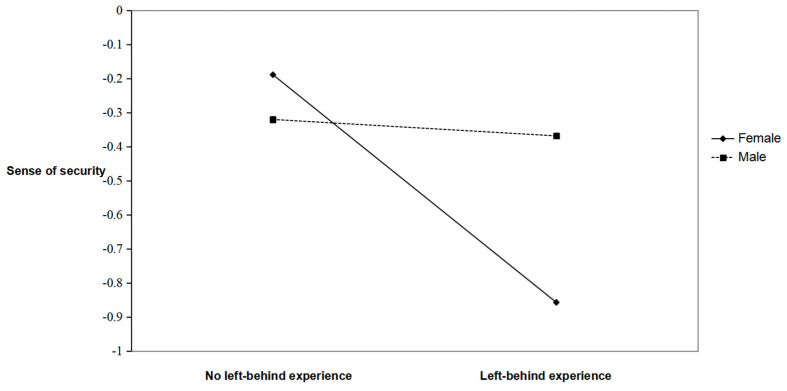
Simple slope diagram of the moderating effect of gender.

**Table 1 behavsci-13-01001-t001:** Descriptive statistics of variables and correlation analysis between variables.

	*M*	*SD*	Left-Behind Experiences	Sense of Security
Left-behind experiences	0.51	0.50		
Sense of security	43.86	11.64	−0.21 ***	
Cyberbullying behavior	19.56	4.22	0.12 ***	−0.09 **

Note: Virtual coding, LBEs (0 for no LBE, 1 for LBE). *** *p* < 0.001, ** *p* < 0.01.

**Table 2 behavsci-13-01001-t002:** Moderated mediation analysis.

Predictors	Model 1			Model 2		
	Sense of Security			Cyberbullying Behavior		
	*SE*	*β*	*t*	*SE*	*β*	*t*
Student cadres	0.06	0.15	2.63 **	0.06	−0.20	−3.31 ***
Left-behind experience	0.06	−0.41	−6.96 ***	0.06	0.20	3.30 ***
Sense of security				0.03	−0.06	−1.99 *
Gender	0.06	−0.14	−2.31 *	0.06	0.27	4.35 ***
Left-behind experience × Gender	0.12	0.62	5.18 ***	0.19	0.13	1.47
*R* ^2^	0.08			0.05		
*F*	23.78 ***			10.40 ***		

Note. * *p* < 0.05, ** *p* < 0.01, *** *p* < 0.001.

**Table 3 behavsci-13-01001-t003:** The mediating effect of sense of security on the relationship between LBEs and CB.

Mediator Variable	Effect	Estimated Effect	Effect Ratio	Boot SE	95% CI
Sense of security	Total effect	0.23 ***		0.06	[0.113, 0.342]
	Direct effect	0.20 ***		0.06	[0.083, 0.321]
	Indirect effect	0.03 *	13.04%	0.01	[0.003, 0.052]

Note. **p* < 0.05, ****p* < 0.001.

**Table 4 behavsci-13-01001-t004:** The mediating effect of gender-specific college students’ sense of security between LBEs and CB.

Mediator Variable	Gender	Effect	SE	95% CI
Sense of security	male college students	0.003	0.006	[−0.007, 0.001]
Sense of security	female college students	0.040	0.020	[0.001, 0.081]

## Data Availability

The corresponding authors have full access to all the data in the study and take responsibility for the integrity of the data and the accuracy of the data analyses.

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
