# Peer review of "Left-Behind Experiences and Cyberbullying Behavior in Chinese College Students: The Mediation of Sense of Security and the Moderation of Gender"

_behavsci, 2023, doi:10.3390/bs13121001_

Round 1

Reviewer 1 Report

Comments and Suggestions for Authors

The study aimed to investigate the correlation between the experience of being left-behind and cyberbullying behavior among Chinese college students, exploring the influence of a sense of security and considering gender as a potential moderator. While the manuscript poses intriguing research inquiries, there seems to be confusion within the authors' treatment of cyberbullying and cyber victimization concepts, particularly evident in Line 54 of the manuscript. The distinction between the two is crucial, as cyberbullying involves the roles of bully, victim, or both, thus cyber victimization represents a separate aspect of this phenomenon.

Moreover, using the Zhou et al. questionnaire to gauge cyberbullying presents an issue. This questionnaire comprises distinct forms for perpetrators and victims of cyberbullying, each consisting of 18 items. However, the authors merged the scores from both the bully and victim subscales to generate a singular score, which may obscure the differentiation between those who solely perpetrate cyberbullying and those who exclusively experience victimization—a significant flaw in the analysis.

To address these methodological concerns, I recommend reevaluating the data separating the victims' and cyberbullying subscales for a more precise analysis. Clarifications regarding the measurement of being left behind (LBE) and other demographic parameters used in the study would also enhance the manuscript's clarity. Including a model diagram could aid in comprehending the proposed associations. Besides, the authors should also justify using model 8 and not 59 in the paper.

Additionally, revisiting the definition of cyberbullying and conducting a thorough review of pertinent literature to include recent references specific to the Chinese context would strengthen the discussion. It's imperative to make sure to correct outdated or unsubstantiated claims with proper references to support the findings.

Furthermore, meticulous proofreading is essential to rectify several typographical errors throughout the manuscript, such as those evident in Lines 40-48, which might be residual comments from previous reviewers.

In summary, while the research question holds promise, a comprehensive revision encompassing a refined analysis and more explicit conceptualization is advisable to enhance the manuscript's scholarly rigor and coherence.

Comments on the Quality of English Language

Minor editing of English language is required.

Author Response

Q1:There seems to be confusion within the authors' treatment of cyberbullying and cyber victimization concepts, particularly evident in Line 54 of the manuscript. The distinction between the two is crucial, as cyberbullying involves the roles of bully, victim, or both, thus cyber victimization represents a separate aspect of this phenomenon.

A1: Thanks for the suggestions on the concept of cyberbullying.

This study mainly discussed the bullying behavior in cyberbullying. The cyberbullying (CB) in the manuscript referred to the role of bully, so the unclear definition in the manuscript ws modified. Authors revised ”Cyberbullying behavior (CB) has emerged and continues to increase as a new form of bullying [9, 10]. CB, also called cyber victimization, is an intentional, repetitive, and aggressive behavior committed by individuals or groups against other individuals using information and communication technologies [11].” to” Cyberbullying (CB) has emerged and continues to increase as a new form of bullying [9, 10], which is an intentional, repetitive, and aggressive behavior committed by individuals or groups against other individuals using information and communication technologies [11].” (Line 45-48)

Q2:The authors merged the scores from both the bully and victim subscales to generate a singular score, which may obscure the differentiation between those who solely perpetrate cyberbullying and those who exclusively experience victimization—a significant flaw in the analysis.

A2:Thanks to the reviewer for the suggestions on measures of cyberbullying.

This study used cyberbullying subscale of Cyberbullying Scale to measure extent of bullying others in the online environment which compiled by Erdur and Kavsut [46] and revised by Zhou et al. The score used is the score of this dimension and not the score of the overall scale. Therefore, the author has further clarified the statement in the manuscription.

The authors revised “The scale includes a cyberbullying subscale and a victim-bullying subscale. The cyberbullying subscale gauge the extent of bullying others in the online environment. The victim-bullying subscale measures the extent of being bullied by others online. The scale contains total of 36 questions, including 18 on the cyberbullying subscale (e.g., ‘I have insulted someone in a chatroom’) and 18 questions on the victim-bullying subscale (e.g., ‘I have been slandered by fake photos of me on the internet’). Participants were asked to report their experiences on 4-point Likert-type items, providing a score for having experienced cyberbullying in the past year (where 1 = ‘it has never happened to me’, 2 = ‘it happened once or twice’, 3 = ‘it happened three to five times’, and 4 = ‘it happened more than five times’). The higher the total score, the higher the level of individual’s involvement in CB. In this study, the scale had good validity (KMO=0.94, p<0.001) and reliability, with a Cronbach’s alpha was 0.92.” to “The scale includes a cyberbullying subscale and a victim-bullying subscale, with 18 items for each. This study used the cyberbullying subscale to measure extent of bullying others in the online environment (e.g., ‘I have insulted someone in a chatroom’). Participants were asked to report their experiences with 4-point Likert-type, providing a score for having experienced cyberbullying in the past year (where 1 = ‘it has never happened to me’, 2 = ‘it happened once or twice’, 3 = ‘it happened three to five times’, and 4 = ‘it happened more than five times’). The higher the score, the higher the level of individuals bullying others online. In this study, the subscale had good validity (KMO=0.94, p<0.001) and reliability, with a Cronbach’s alpha was 0.92.” (Line 149-158)

Q3:Clarifications regarding the measurement of being left behind (LBE) and other demographic parameters used in the study would also enhance the manuscript's clarity. Including a model diagram could aid in comprehending the proposed associations. Besides, the authors should also justify using model 8 and not 59 in the paper.

A3:Thanks for the advice on data analysis methods used in this study.

Measurement of demographic variables and LBEs:

For the measurement of LBEs and other demographic variables, the demographic variables used in this study were student cadre and gender. The results found significant differences in CB in terms of student cadre and gender. The definition of left-behind college students used in this study is ”College students with LBEs are those who are enrolled in school with one or both parents working away from home for over six months before the students turned 16 [3]. (Line 33)”. Therefore, the question “Did one or both of your parents go out to work for more than six months from your birth to your 16 years old? (0 = no, 1 = yes)” was used to measure LBEs in this study. The authors had specifically added the measurement of LEBs in the manuscript: “LBEs was evaluated using a questionnaire designed by the authors that included gender, student cadre and LBE. According to the definition of LBEs of Zuo et al. [3], the question of LBEs was “Did one or both of your parents go out to work for more than six months from your birth to your 16 years old? (0 = no, 1 = yes)”. (Line 160-163)

Model Diagram:

Based on the General Aggression Model and attachment theory, combined with existing empirical research, the study explored the relationship between LBEs and CB among college students, as well as the mediating role of sense of security and moderating role of gender between the two. This study hypothesizes that ”1) college students’ LBEs positively associated with CB; 2) sense of security mediates the relationship between college students’ LBEs and CB; and 3) gender moderates the relationships between college students’ LBEs and sense of security and between LBEs and CB”. Therefore, the authors added the model diagram, "Figure 1. The moderated mediation effect among LBEs, CB, sense of security and gender.” (Line 133)

The reason of choosing Model 8 for analysis:

Since one of the prerequisites of the moderating effect is that the moderator has no effect on the independent variable, if the moderator has an effect on the independent variable, there is a nonlinear relationship between the independent variable and the dependent variable. Therefore, gender could not be used as a moderating variable between sense of security and CB in this study, which was ultimately validated using Model 8.

Q4:Revisiting the definition of cyberbullying and conducting a thorough review of pertinent literature to include recent references specific to the Chinese context would strengthen the discussion. It's imperative to make sure to correct outdated or unsubstantiated claims with proper references to support the findings.

A4:Thanks very much for the suggestions of definition of CB and related references.

Regarding the definition of CB, this study mainly focuses on bullying others in online environment and defined it as “Cyberbullying (CB) has emerged and continues to increase as a new form of bullying [9, 10], which is an intentional, repetitive, and aggressive behavior committed by individuals or groups against other individuals using information and communication technologies [11].” (Line 45-48)

The authors supplemented the references accordingly.

Their higher involvement in CB may be related to the high work pressure and strong competitiveness as a student cadre [55, 56]. Confronted with the questioning and misunderstanding from their classmates, they are more inclined to express their emotions, such as sadness and anger, in the online environment and cause harm to others [57]. It also reminds us to be more attentive towards student cadres’ emotional and psychological health. (Line 277-282)

This outcome supports Hypothesis 1 and points to an important link between college students’ LBEs and CB. The family function of LBC, whose parents left home to work, is relatively weak [58]; family cohesion and parental support are weak, and their children perceive less family warmth [59, 60]. (Line 287-291)

Depressed emotions in real life are brought into the unfettered network environment, thereby facilitating CB. In addition, the positive parenting style of college students with LBEs is significantly lower than that of college students without LBEs [59, 60]. (Line 292-295)

Low-income family structure, negative parenting style, and less parent-child interaction may lead to CB [61]. (Line 298)

College students with LBEs usually experience low-income family functioning in childhood, and they display poor communication among family members, loose emotional connections, and inadequate security attachment [61, 67]. (Line 340-343)

This may be because the family environment—when separated from their parents—causes females to be more sensitive [68, 69]. Female gender characteristics make them exhibit a higher demand for a stable living environment and parental support [70]. The lack of parental presence and care make females more susceptible to unsafe attachments, thereby reducing their sense of security [68, 69]. (Line 352-356)

Added to the list of references (Line 548-562, 576-585):

  1. Yang, J.; Li, S.; Gao, L.; Wang, X. Longitudinal associations among peer pressure, moral disengagement and cyberbullying perpetration in adolescents. Computers in Human Behavior.2022, 137, 107420. doi: 1016/j.chb.2022.107420 (Line 548-549)
  2. Rao, S.S. The application of "four power separation" of class cadres in class management". Ability and Wisdom, 2020, 16, 216-17. (Line 550)
  3. Song, N. Research on the existing problems and countermeasures of the primary school class cadre system. Master Degree Thesis. Ludong University. 2017, https://kns.cnki.net/KCMS/detail/detail.aspx?dbname=CMFD201801&filename=1017827454.nh (Line 551-553)
  4. Zhang, X.; Fu, L.P. The relationship between family functioning and negative affect of college students with Lift-behind experience: the chain mediating role of self-esteem and intolerance of uncertainty. Journal of Chengdu Normal University, 2023, 39(7), 48–58 (Line 554-556)
  5. Lan, X.; Wang, W. Is early left-behind experience harmful to prosocial behavior of emerging adult? The role of parental autonomy support and mindfulness. Current Psychology, 2022, 41(4), 1842– https://doi.org/10.1007/s12144-020-00706-3(Line 557-558)
  6. Zhang, C.; Yang, X.; Xu, W. Parenting style and aggression in Chinese undergraduates with left-behind experience: The mediating role of inferiority. Children and Youth Services Review, 2021, 126, 106011. doi: 10.1016/j.childyouth.2021.106011(Line 559-560)
  7. Zhang, S.; Zhang, Y. On the relationship among parent-child attachment, the tendency of shame and cyberbullying in rural senior school students. Journal of Hangzhou Normal University (Natural Science Edition). 2020, 19(2), 139-144+171. (Line 561-562)
  8. Xiao, Z.L.; Li, Y.H.; Xiao, K.J. Research on the current situation and countermeasures of parent-child relationship among college students with Left-behind experience in rural areas: Based on a survey of post-00s college students in Guangdong Province. Journal of Qingyuan Polytechnic. 2023, 16(1), 83–90 (Line 576-578)
  9. Wan, J.J.; Ji, L.L.; Wu, L.N.; Zhang, Y.F.; Liu, G.G.; Gu, H.; Zhao, J.F. Relationship between parent-child cohesion and sense of security in rural left-behind junior students: a study with cross-lagged modeling. Studies of Psychology and Behavior, 2021, 19(4), 500–506. (Line 579-581)
  10. Chang, S.M.; Song, Y.S.; Guo, H. Home Chaos and Migrant Children’s Security: the Moderating Effect of Gender. Chinese Journal of Special Education. 2016, 4, 66-70+78. (Line 582-583)
  11. Gordon, E.; Lee-Koo, K. Addressing the security needs of adolescent girls in protracted crises: Inclusive, responsive, and effective? Contemporary Security Policy. 2021, 42(1), 53–82. https://doi.org/10.1080/13523260.2020.1826149 (Line 584-585)

Q5:Meticulous proofreading is essential to rectify several typographical errors throughout the manuscript, such as those evident in Lines 40-48, which might be residual comments from previous reviewers.

A5:Thanks to the suggestions on the typographical errors, the authors conducted a detailed review and revised accordingly.

The authors deleted “The introduction should briefly place the study in a broad context and highlight why it is important. It should define the purpose of the work and its significance. The current state of the research field should be carefully reviewed, and key publications cited. Please highlight controversial and diverging hypotheses when necessary. Finally, briefly mention the main aim of the work and highlight the principal conclusions. As far as possible, please keep the introduction comprehensible to scientists outside your particular field of research. References should be numbered in order of appearance and indicated by a numeral or numerals in square brackets—e.g., [1] or [2,3], or [4–6]. See the end of the document for further details on references.

Reviewer 2 Report

Comments and Suggestions for Authors

The abstract is concise in a single paragraph with less than 200 words. You have an overview of the study. The main question is addressed in a broad context, highlighting the objective of the research. Regarding the methods, the mediation of security and gender moderation is indicated, as well as the use of questionnaires to collect the evaluations of university students with and without abandonment experiences. The results are specified in summary form to replicate the findings and main conclusions. Keywords are relevant to the topic.

The introduction begins by explaining the family context and economic growth in China from the 1980s onwards, highlighting the consequences of abandoned university students LBCs and LBEs. A paragraph appears from the template referring to the introduction that is not part of the study of lines 40 to 48 and that must be removed.

In the introduction I consider that key and interesting publications are cited, to indicate background and see the current state of the topic of study. Three hypotheses are clearly specified. The explanation and writing are clear and understandable. References are numbered as indicated. I consider that in this introduction you should briefly mention the main objective sought and highlight the main conclusions.

Regarding the method, the number of participants is indicated by sex, university, average age, recovery rate and health. It was approved by the Faculty's Academic Ethics Committee. The Cyberbullying Scale indicates validity and reliability. It explains that the Sense of Security Questionnaire is used to measure the individual's sense of security. It has good reliability and validity, which is why it has been widely applied. They use the Likert scale of four and five points respectively. The possible exclusion criteria must be clarified, since as there are many questionnaires, erroneous or inconclusive answers may have been produced. The analysis tools are specified with the data clear enough to be replicated. I consider that the procedure followed with the method and analysis instruments needs to be given a little more detail. It is not specified if there were any restrictions on the availability of materials and tools for the analysis.

The results are appropriately divided into subtitles. Thus, the test and variance control of the common method, the demographic characteristics of university students in relation to CB according to gender differences, the analysis of differences in CB and feeling of security among university students with LBE are explained. In short, the description of the results and their interpretation are presented precisely, enabling discussion and conclusions to be derived. The tables and figures shown clearly correspond to the variables and interests of the study.

The discussion focuses on the results presented and the interpretation produced as strong points of the study. In an orderly manner, the proposed variables that have been adequately linked and verified with the three hypotheses are discussed. The findings of this study support the conclusion with clear references from the gender perspective and the attention that must be paid to the emotional and psychological health of students.

Limitations and future lines of research are presented. I positively value the self-criticism described, as well as the determination of future research based on the prospective carried out.

Although the discussion is detailed and sufficient, the authors require conclusions adjusted to the main objective and, also, to the hypotheses and variables of the study. Two quotes from co-author Weichen Wang appear in the references.

Author Response

Q1:A paragraph appears from the template referring to the introduction that is not part of the study of lines 40 to 48 and that must be removed.

A1:Thanks to the suggestions on the manuscript Line 40-48, which the authors have revised.

The authors deleted “The introduction should briefly place the study in a broad context and highlight why it is important. It should define the purpose of the work and its significance. The current state of the research field should be carefully reviewed and key publications cited. Please highlight controversial and diverging hypotheses when necessary. Finally, briefly mention the main aim of the work and highlight the principal conclusions. As far as possible, please keep the introduction comprehensible to scientists outside your particular field of research. References should be numbered in order of appearance and indicated by a numeral or numerals in square brackets—e.g., [1] or [2,3], or [4–6]. See the end of the document for further details on references.

Q2:I consider that in this introduction you should briefly mention the main objective sought and highlight the main conclusions.

A2:Thanks for the advice of the part of Introduction.

The main purpose of this study is to explore the long-term adverse effects of LBE on individuals and its relationship with CB, therefore, this study explored the mediating role of security and the moderating role of gender between the two based on the General Aggression Model and attachment theory.

The authors revised “The current study adopted a cross-sectional approach to explore the relationship between LBEs and CB among college students and examined the mediating effect of sense of security and the moderating effect of gender. Based on General Aggression Model and attachment theory, this study hypothesises that 1) college students’ LBEs positively associated with CB; 2) sense of security mediates the relationship between college students’ LBEs and CB; and 3) gender moderates the relationships between college students’ LBEs and sense of security and between LBEs and CB.” to “In order to investigate the long-term adverse effects of LBEs and how it affects college students’ CB, this study adopted a cross-sectional approach and based on General Aggression Model and attachment theory to explore the relationship between LBEs and CB, as well as the mediating effect of sense of security and the moderating effect of gender. This study hypothesizes that 1) college students’ LBEs positively associated with CB; 2) sense of security mediates the relationship between college students’ LBEs and CB; and 3) gender moderates the relationships between college students’ LBEs and sense of security and between LBEs and CB (Figure 1).” (Line124-133)

The main conclusions of this study were reflected in the hypotheses, specifically “This study hypothesizes that 1) college students’ LBEs positively associated with CB; 2) sense of security mediates the relationship between college students’ LBEs and CB; and 3) gender moderates the relationships between college students’ LBEs and sense of security and between LBEs and CB (Figure 1).” (Line 128-131)

Q3:The possible exclusion criteria must be clarified, since as there are many questionnaires, erroneous or inconclusive answers may have been produced. & I consider that the procedure followed with the method and analysis instruments needs to be given a little more detail.

A3:Thanks to the reviewer for the questions on the criteria for the exclusion and the suggestions in the research methods section.

The questionnaire is anonymous and there is no right or wrong answer, so the final criterion for excluding participants is regular answers. In response to this problem, the authors provided a supplementary description in section 2.4 that revised “Undergraduates from multiple universities agreed to participate in the online survey.” to “Undergraduates from multiple universities agreed to participate in the online survey and the online survey could be submitted only if all questions were completed.” (Line 182)

The questionnaires used in this study had good reliability and validity among college students. According to the suggestions, the authors revised this section in two ways.

First, the authors clarified the description of the measurement of cyberbullying. Revised “The scale includes a cyberbullying subscale and a victim-bullying subscale. The cyberbullying subscale gauge the extent of bullying others in the online environment. The victim-bullying subscale measures the extent of being bullied by others online. The scale contains total of 36 questions, including 18 on the cyberbullying subscale (e.g., ‘I have insulted someone in a chatroom’) and 18 questions on the victim-bullying subscale (e.g., ‘I have been slandered by fake photos of me on the internet’). Participants were asked to report their experiences on 4-point Likert-type items, providing a score for having experienced cyberbullying in the past year (where 1 = ‘it has never happened to me’, 2 = ‘it happened once or twice’, 3 = ‘it happened three to five times’, and 4 = ‘it happened more than five times’). The higher the total score, the higher the level of individual’s involvement in CB. In this study, the scale had good validity (KMO=0.94, p<0.001) and reliability, with a Cronbach’s alpha was 0.92.” to “The scale includes a cyberbullying subscale and a victim-bullying subscale, with 18 items for each. This study used the cyberbullying subscale to measure extent of bullying others in the online environment (e.g., ‘I have insulted someone in a chatroom’). Participants were asked to report their experiences with 4-point Likert-type, providing a score for having experienced cyberbullying in the past year (where 1 = ‘it has never happened to me’, 2 = ‘it happened once or twice’, 3 = ‘it happened three to five times’, and 4 = ‘it happened more than five times’). The higher the score, the higher the level of individuals bullying others online. In this study, the subscale had good validity (KMO=0.94, p<0.001) and reliability, with a Cronbach’s alpha was 0.92.” (Line 149-158)

Second, the authors added a measure of LBE. Added “LBEs was evaluated using a questionnaire designed by the authors that included gender, student cadre and LBE. According to the definition of LBEs of Zuo et al. [3], the question of LBEs was “Did one or both of your parents go out to work for more than six months from your birth to your 16 years old? (0 = no, 1 = yes)”. (Line 160-163)

Q4:Although the discussion is detailed and sufficient, the authors require conclusions adjusted to the main objective and, also, to the hypotheses and variables of the study. 

A4:Thanks very much for the suggestions of the section of conclusion.

According to the suggestions, the authors revised “This study found that college students with LBEs had significantly higher CBs than those without LBEs. The sense of security partially mediated the relationship between college students' LBEs and CB. Additionally, gender moderated the mediating effect of sense of security between LBEs and CB. Compared with male college students, the mediating effect was significant only among female college students. Having LBEs is a common phenomenon among Chinese college students. To reduce the level of CB in college students, future researchers and educators should pay more attention to the feelings of inferiority experienced by college students, especially those with LBEs.” to “This study found that 46.47% of college students had LBEs, indicating that Having LBEs is a common phenomenon among Chinese college students. LBEs was an environmental factor that influences CBs, and the relationship between the two was mediated by a sense of security, and gender mediated the relationship between the two. Compared with male college students, the mediating effect was significant only among female college students. In the future, researchers and teachers should pay more attention to college students' sense of security, so as to reduce the incidence of CB among college students and alleviate the adverse effects of LBEs on college students.”. (Line 401-408)

Reviewer 3 Report

Comments and Suggestions for Authors

The manuscript is interesting, however, it needs a major revision.

What is this? Lines 40-48: The introduction should briefly place the study in a broad context and highlight why 40 it is important. It should define the purpose of the work and its significance. The current 41 state of the research field should be carefully reviewed and key publications cited. Please highlight controversial and diverging hypotheses when necessary. Finally, briefly mention the main aim of the work and highlight the principal conclusions. As far as possible, please keep the introduction comprehensible to scientists outside your particular field of research. References should be numbered in order of appearance and indicated by a numeral or numerals in square brackets—e.g., [1] or [2,3], or [4–6]. See the end of the document for further details on references.

There is no info on how the Left-Behind Experiences were measured. This is a crucial variable for the study and authors even report M and SD for LBE but there is no info on the measure.

Author Response

Q1: What are Lines 41-48?

A1: Thanks to the suggestions on the manuscript Line 40-48, which the authors have revised. 

The authors deleted "The introduction should briefly place the study in a broad context and highlight why it is important. It should define the purpose of the work and its significance. The current state of the research field should be carefully reviewed, and key publications cited. Please highlight controversial and diverging hypotheses when necessary. Finally, briefly mention the main aim of the work and highlight the principal conclusions. As far as possible, please keep the introduction comprehensible to scientists outside your particular field of research. References should be numbered in order of appearance and indicated by a numeral or numerals in square brackets—e.g., [1] or [2,3], or [4–6]. See the end of the document for further details on references".

Q2: There is no info on how the Left-Behind Experiences were measured. This is a crucial variable for the study and authors even report M and SD for LBE but there is no info on the measure.

A2: Thanks very much for the suggestions of the measurement of LBEs.

This study defined the LBEs as “College students with LBEs are those who are enrolled in school with one or both parents working away from home for over six months before the students turned 16 [3].” (Line33-35). Therefore, the question “Did one or both of your parents go out to work for more than six months from your birth to your 16 years old? (0 = no, 1 = yes)” was used to measure LBEs in this study.

The authors had specifically added the measurement of LEBs in the manuscript: “LBEs was evaluated using a questionnaire designed by the authors that included gender, student cadre and LBE. According to the definition of LBEs of Zuo et al. [3], the question of LBEs was “Did one or both of your parents go out to work for more than six months from your birth to your 16 years old? (0 = no, 1 = yes)”. (Line 160-163)

Round 2

Reviewer 1 Report

Comments and Suggestions for Authors

The authors have satisfactorily addressed all the queries. The paper is now fit for publication as it stands.

Comments on the Quality of English Language

Minor editing of English is required

Reviewer 3 Report

Comments and Suggestions for Authors

The authors improved the manuscript, it is aacceptable.